# An Evaluation of the Hand Hygiene Behaviour and Compliance of the General Public When Using Public Restrooms in Northern Ireland (NI) during the Initial Weeks of the Novel Coronavirus (COVID-19) Pandemic

**DOI:** 10.3390/ijerph18126385

**Published:** 2021-06-12

**Authors:** Aaron Lawson, Robert Cameron, Marie Vaganay-Miller

**Affiliations:** Belfast School of Architecture and the Built Environment, Ulster University, Newtownabbey BT37 0QB, UK; rj.cameron@ulster.ac.uk (R.C.); m.vaganaymiller@ulster.ac.uk (M.V.-M.)

**Keywords:** hand hygiene, COVID-19, behaviour, compliance, observation, general public

## Abstract

Background: The ongoing novel coronavirus (COVID-19) global pandemic has resulted in significant levels of morbidity and mortality worldwide, particularly among the elderly and immuno-suppressed groups. Although adequate hand hygiene (HH) behaviour and compliance is widely accepted as being the most effective self-protective measure in preventing the spread of diseases like COVID-19, previous research suggests that normal hand hygiene compliance is poor, but generally improves during a disease pandemic. This research aimed to evaluate the hand hygiene behaviour and compliance of the general public in the initial weeks of the COVID-19 pandemic in Northern Ireland (NI). Methods: This cross-sectional study involved the use of infrared-imaging cameras to observe the hand hygiene behaviour and compliance of the general public when using one set of male and female public restrooms. Results: The findings of this study indicated that the level of hand hygiene compliance of the general public was poor in the initial weeks, with 82.93% overall not washing their hands adequately. Conclusions: Inadequate HH behaviour and compliance may have added significantly to the rapid rate of spread of COVID-19 in the initial weeks of the pandemic in NI. Current public health campaigns do not appear, based on this study, to have the desired impact and may need to be reviewed or re-enforced in order to achieve the levels of hand hygiene compliance required to slow the spread of COVID-19 and other communicable diseases in the future.

## 1. Introduction

The ongoing novel coronavirus (designation COVID-19) outbreak which originated in Wuhan, China in December 2019 rapidly developed into a global pandemic in early 2020 [1,2,3], resulting in significant levels of morbidity and mortality especially amongst the elderly and immuno-suppressed groups. At the time of writing, there have been around 141,057,106 cases in 117 countries, and 3,015,043 deaths worldwide [4].

Adequate hand hygiene behaviour and compliance are recognised as being the most effective method of preventing communicable disease transmission, including COVID-19 [5,6,7]. The purpose of adequate hand hygiene behaviour is to remove any potentially pathogenic microorganisms from the hands and minimise any potential transmission from person to person or from contaminated physical surfaces. Adequate hand hygiene behaviour involves washing hands with warm water and soap and scrubbing thoroughly, and this definition is based upon previous research findings as well as national guidelines such as the World Health Organisation’s ‘5 Moments for Hand Hygiene’ technique, which is a required infection prevention routine in health care settings [8,9,10].

Likewise, adequate hand drying is important in preventing the transmission of communicable pathogens like COVID-19, as it helps removes any remaining communicable pathogens left in wet areas of the hands, therefore minimising the further opportunity for person-to-person transmission [11]. The length of time spent washing and drying hands is also important for reducing the pathogenic loading of hands, with current guidelines recommending a minimum of twenty seconds for each process [12,13].

Most previous studies which have examined the hand hygiene behaviour and compliance of the general public have found compliance to be varied, but mostly poor [14,15,16,17,18]. For instance, Anderson and colleagues [14] established a hand hygiene compliance rate of 58% for adult members of the general public at a petting zoo, and Freeman et al. [15] estimated that only 19% of the global population wash their hands with water and soap. Hand hygiene compliance rates for children are also generally poor, with van Beeck et al. [16] reporting an overall compliance rate of 31% in their study. In another study by Lary et al. [19], children’s baseline hand hygiene compliance was established as being around 19%. Gender differences in some previous studies also found that females generally have better hand hygiene compliance rates compared to males [20,21].

The reasons for this are complex, but personal health is often considered to be low on people’s list of motives, and hand hygiene is considered to be an inherent behaviour, often being established at an early age meaning that hands are often only washed when visibly dirty or to make them look or smell good [22,23]. Therefore, most instances of inadequate hand hygiene are likely due to forgetfulness, rather than intention, and can be difficult to change as time progresses [23,24].

Only when there is an immediate and highly publicised threat to personal health, such as during a disease pandemic or epidemic, does hand hygiene behaviour improve for the better [25]. Other, previously conducted studies on hand hygiene behaviour and compliance during different communicable disease outbreaks reaffirm this point, suggesting that people’s hand hygiene behaviour and compliance generally improve in response to a disease outbreak [25,26]. For instance, Meilicke et al. [26] found an increase in the self-reported rate of good hand hygiene practice amongst members of the general public in Germany from 50.90% in 2008 to 61.10% in 2009 in response to the novel influenza A H1N1 pandemic. Similarly, Park et al. [27] found that people’s frequency of hand hygiene practice increased by 30.30% compared to the previous year after the H1N1 pandemic. A recent survey commissioned on behalf of the UK government by Imperial College London [28] stated that 83% of members of the general public washed their hands more frequently now than before the COVID-19 pandemic. Despite this, other studies suggest that following a disease epidemic or pandemic, adequate hand hygiene behaviour and compliance discontinue over time due to the perceived lack of threat from a disease reoccurrence [29,30].

The common theme with many of these previous studies however is that they rely on using self-reporting methods when determining hand hygiene behaviour and compliance. The major issue with self-reporting methods is that they are often not a reliable indicator of true behaviour compared to other methods such as observation (17–18). Most of the research conducted during the course of the COVID-19 pandemic has relied on using self-reporting methods [31,32,33], many of which have focused solely on the availability of handwashing facilities, rather than the efficacy of hand hygiene behaviour within the general population.

If adequate hand hygiene behaviour and compliance are not maintained at key times, like after using the toilet, before handling or preparing food, and after being outside or in close contact with others, then this will facilitate the transmission of communicable diseases like COVID-19 more easily within the general public and place more pressure on national healthcare services in coping with the ongoing pandemic. It is therefore essential that the general public follow government and public health advice and adhere to personal protective measures, most important of which include adequate hand hygiene behaviour and compliance [5,6,7].

The aim of this research was to establish the level of hand hygiene behaviour and compliance of the general public during the initial weeks of the novel coronavirus (COVID-19) pandemic in a public restroom setting in Northern Ireland (NI).

## 2. Materials and Methods

To fulfil the aim of the study, an observational, cross-sectional study design was adopted to establish the level of hand hygiene behaviour and compliance of the general public when using public restrooms during the initial weeks of the COVID-19 pandemic. This included one set of male and female public restrooms which were located in Belfast, NI. Indirect observation was undertaken at each public restroom between 10–20 March 2020. Public restrooms located in Belfast city centre were selected for their convenience, and also because it was theorised that as a closed space; they could provide ample opportunity for the spread of COVID-19 between members of the general public after using the toilet.

### 2.1. Patient and Public Involvement

Members of the general public who voluntarily used the public restrooms under observation were included as participants in this research. Voluntary consent to participate was achieved via the use of signage erected two weeks prior to live observation and was placed on the exterior doors of each public restroom informing research subjects that important public health research was being conducted and that no one could be identified. This was done in consideration of the Hawthorne Effect, which is a type of reactivity when people are aware they are being observed [34] and to allow research subjects to become accustomed to the presence of the cameras in each restroom before live observation was due to take place. Belfast is also the largest city in Northern Ireland with a population of around 342,000 people [35], and it was theorised that voluntary participation would be facilitated to a much greater extent compared to other locations.

Both sets of restrooms only had liquid soap available for washing hands, and Dyson hand dryers for drying hands. There were 3 cubicles, 5 urinals, 3 sinks and 2 hand dryers in the male restroom. In the female restroom, there were 4 cubicles, 4 sinks and 2 hand dryers.

Indirect observation of the general public’s hand hygiene behaviour and compliance was performed using infrared-imaging cameras based upon a similar methodology employed in a recent, previous study [36]. This meant that there was no live observer present during the quantitative data collection process. Infrared-imaging cameras were the preferred option as they helped ensure research subject anonymity as per ethical considerations. Figure 1 below is an image taken from one of the public restrooms under observation in this research.

The infrared-imaging cameras used in this study were positioned above the entrances within each public restroom above head height to remain discreet. Only the wash-hand basin and hand dryer areas in each restroom could be observed. Live observation in each public restroom occurred between the times of 07:00 and 21:00, Monday to Sunday. For the purpose of this study, this is known as the observation period. Due to the use of thermal imaging cameras for observation, it was only possible to collect data on research subject demographics by gender (male or female), and adult or child. It was also not possible to identify the number of research subjects who did not consider using the public restrooms under observation after reading the signage on the exterior doors.

The data collected (infrared-video footage) was pre-coded for analysis using pre-determined criteria that recorded the relevant hand hygiene behaviours of each individual research subject and included:Hand hygiene intention (did the person go to wash their hands?)Hand hygiene compliance (how did they wash their hands? Did they wash using only water or water and soap?)Time spent washing hands (was it </>20 s?)Hand drying intention (did the person go to dry their hands after washing?)Hand drying compliance (which method of drying did they use to dry their hands?)Time spent drying hands (was it </>15 s? This figure was based upon Dyson guidelines for their hand dryers).Gender (male or female)?Adult or child?

These behaviours were used to establish research subjects’ level of hand hygiene compliance into four categories relevant to the context of this study which were based upon a recent, previous study [36]. The categories included:Adequate hand hygiene: this involved washing hands with water, soap and then lathering for twenty seconds or more and scrubbing in various rotations and interlocking of fingers, after which hands are rinsed with water to remove soap excess and then dried properly using an appropriate drying method (only a Dyson hand dryer was available in this study) for fifteen seconds or more also [36].Basic hand hygiene: this involved washing hands with water, soap and scrubbing hands in various rotations and interlocking of fingers after which hands are rinsed with water to remove soap excess, and then dried afterward using an appropriate drying method (Dyson hand dryer) but not for the recommended minimum length of time [36].Poor hand hygiene: this involved any other combination of steps not fitting the two previous categories [36].Non-hand hygiene: this involved not washing or drying hands at all.

Two observers were used to analyse the recorded thermal footage, with there being a third observer to verify both observers’ analysis during the study. This was done to minimise any observation bias and for consistency [36].

### 2.2. Data Analysis

The pre-coded data collected was input into, and analysed using IBM’s SPSS Statistical Software (v.24) (IBM, New York, NY, USA). Both descriptive and inferential statistics were performed. Chi-square analysis was used to identify statistically significant comparisons between relevant variables. The accepted statistical significance level was determined as *p* ≤ 0.05, with Confidence Levels of 95% (CI) reported where applicable.

## 3. Results

In total, 498 research subjects (members of the general public) were observed using both sets of public restrooms under observation. This included 254 males and 244 females, of which 453 were adults and 45 were children. Only those who knowingly used the restrooms under observation and were clearly present were included in the final study results. It was not possible to determine how many research subjects chose not to use the restrooms during the observation period due to methodological constraints and ethical considerations. A summary of the hand hygiene compliance rate of research subjects in this study is shown in Table 1 below.

Around 82.93% of research subjects overall practiced inadequate hand hygiene compliance, and just 17.07% practiced adequate compliance.

Gender differences revealed that females were more likely to practice basic hand hygiene (68.03%) compared to males (57.48%) which was statistically significant (χ^2^ = 5.92, *p* = 0.02). Additionally, significantly more males (9.84%) practiced non-hand hygiene compliance compared to females (4.51%) (χ^2^ = 5.28, *p* = 0.02).

Equally, differences between adult and child hand hygiene behaviour and compliance revealed that only 14.35% of adults practiced adequate hand hygiene compared to 44.44% of children which was highly statistically significant (χ^2^ = 26.19, *p* = < 0.01). Likewise, 64.02% of adults and 44.89% of children practiced basic hand hygiene which was also statistically significant (χ^2^ = 4.00, *p* = 0.05). About 7.95% of adults practiced non-hand hygiene compared to zero children which was also statistically significant (χ^2^ = 3.86, *p* = 0.05).

The average (mean) duration of time spent washing hands for all research subjects was 18.66 s (Std. dev: 13.91), and 13.46 s for hand drying (Std. dev: 9.49). There were no statistically significant differences between males and females regarding the time spent washing and drying. For adults versus children, the average duration spent washing hands for adults was 17.17 s (SD: 12.08), and 32.51 s for children (SD: 20.75). Regarding hand drying, the average duration of time spent by adults was 12.48 s (SD: 8.16), and 22.26 s for children (SD: 14.85). Figure 2 below shows this information.

## 4. Discussion

Despite adequate hand hygiene behaviour and compliance being documented as being the best self-protective measure against the spread of COVID-19 and other communicable diseases, the findings of this study are highly indicative that most members of the general public during the initial weeks of the COVID-19 pandemic in NI were washing their hands inadequately. The level of adequate hand hygiene compliance found in this study was less than other previously conducted studies during disease outbreaks [19,22,23]. This implies that while disease outbreaks like COVID-19 usually generate an improvement in overall intention to practice hand hygiene compliance compared to pre-pandemic circumstances, there still remained a large proportion of the general public who did not wash their hands adequately enough even during an ongoing disease pandemic. The implications of this finding are profound in terms of evaluating how the public behaves with respect to the prevention of COVID-19 and other similar communicable diseases. It suggests that the public health communication strategy at the time was not having the desired outcome in terms of achieving adequate hand hygiene behaviour and compliance amongst the general public. As a result, it is theorised that the poor level of hand hygiene behaviour and compliance found may have facilitated the faster transmission of the disease amongst the public at this time. The reasons for the level of inadequate hand hygiene are complex and are potentially due to various factors. For instance, it may be because hand hygiene is often considered an inherent behaviour as in it is ingrained in most people as young children as suggested in previous studies [23]. Therefore, if a person is not taught the adequate method of hand hygiene compliance at an early age, then they will never practice this method as time progresses and they get older, and the poor behaviour is repeated unless they are otherwise corrected. Additionally, it may be because hand hygiene compliance is low on most people’s list of priorities for personal health, and unless hands are visibly soiled, they do not wash their hands at key times such as after using the toilet as has been previously suggested [24]. For those members of the general public who did not wash or dry their hands at all, this may be the result of a lack of knowledge on the importance of adequate hand hygiene behaviour and compliance in preventing the transmission of communicable diseases from person to person. This may indicate that they do not associate the risk of not washing or drying hands with the spread of communicable diseases, or because most people have a generally good level of health, they perceive the risk of contracting COVID-19 as being low. Equally, people who were using the public restrooms in March 2020 under observation in this study might have been less concerned about their hygiene behaviour compared with the rest of the general population.

Additionally, most research on hand hygiene uses self-reporting methods such as questionnaires and surveys to determine hand hygiene behaviour and compliance [13,19]. These are typically not as reliable as observational methods [17,18], because observational data is quantitative, and therefore there it minimises the opportunity for self-reporting bias [13,14]. The use of thermal imaging cameras in this study was evidence of this as previous research has shown [33]. This may account for the difference in the level of hand hygiene compliance established in this study versus previous studies using self-reporting methods [24,25].

The key factor in defining adequate hand hygiene behaviour and compliance is timing. The length of time spent washing and drying hands is important for ensuring that all potential pathogenic microorganisms like COVID-19 are removed. The fact that most members of the general public did not spend a minimum of 20-s washing and drying hands, despite the various public health campaigns at the time, may simply be because most people are poor at estimating the length of time they have spent doing so rather than being a conscious decision on their part. This theory may be supported by the high numbers of people in this study observed practicing basic hand hygiene in both restrooms which is indicative of a clear intent on their part to knowingly and willingly practice hand hygiene after using the toilet. Therefore, this group should be deemed as being a key target for public health intervention designed to improve hand hygiene behaviour and compliance further, and this may be necessary during present times and in future as many of the self-protective behaviours like hand hygiene, the wearing of facemasks and social distancing will continue to be encouraged in the coming months and years as we ease out of national lockdown restrictions.

While other interventions to improve the duration of time people spend washing and drying hands after using the toilet need investigating, a simple solution like a visual or audible timer placed in public restrooms may help achieve this. Similarly, regarding individual hand drying behaviour, most hand dryers are currently limited to a pre-determined time setting (e.g., for 15 s or less), and therefore most people who do not dry their hands adequately are potentially doing so because they trust the time limit set by the hand dryer, rather than current guideline advice. Again, a simple visual or audible timer can address this challenge.

It must also be noted that other, more recent interventions such as the development of the COVID-19 vaccine should not be considered as being able to eliminate the need for hand hygiene going forward as further research is needed to investigate its long-term impacts [37,38].

Gender differences in this study revealed that females had better intention to practice hand hygiene compared to males as shown by the level of basic hand hygiene. This may either be because females inherently practice hand hygiene more often than males due to the influence of key role models like parents and guardians, and that women are generally more compliant as has been found in similar, previous studies [20,21]. This may also account for the reasons why males were observed as being significantly more likely to practice non-hand hygiene compliance in this study. Conversely, males that did wash their hands spent on average longer washing and drying their hands than females, and overall, males washed their hands more efficiently than females.

In addition, the findings of this study suggested that children are significantly better at practicing adequate hand hygiene compliance compared to adults. This may be due to the emphasis placed on them at a young age by key role models like teachers and parents in washing their hands adequately. This implies that children may be useful for promoting the message of adequate hand hygiene behaviour and compliance in a future public health communication campaign, acting as good role models for not just other children, but adults as well. Particularly those adults who frequently practice inadequate hand hygiene compliance. Additionally, children on average spent longer washing and drying their hands compared to adults. This may be because children are often instructed by adults to do so, and therefore do not question the instruction. It may also be because external barriers or influences on hand hygiene compliance such as time pressure, lack of available wash hand basins or hand dryers, or the presence of other people do not affect their behaviour to the same extent as adults. Exploring children’s motivation for practicing adequate hand hygiene compliance may help explain the differences in behaviour between adults and children and help improve adult compliance.

Due to ethical, time and resource constraints, only one set of male and female public restrooms could be observed which was one limitation of this research worth mentioning. Additionally, there may have been an element of selection bias in the recruitment of research subjects as only members of the general public who felt comfortable using the public restrooms under observation were included in the study and classified as representing the general public.

## 5. Conclusions

This study aimed to establish the hand hygiene behaviours and level of compliance of the general public during the initial weeks of the novel coronavirus (COVID-19) pandemic in a public restroom setting in NI. The findings indicated that despite government and public health advice at the time on the importance of adhering to self-protective measures like practicing adequate hand hygiene behaviour and compliance to prevent the spread of COVID-19, most of the general public continued to wash their hands inadequately. This suggests that the public health campaigns at the time did not appear to have the desired impact in changing people’s behaviour regarding hand hygiene adequacy and compliance and may have added significantly to the speed of transmission of COVID-19 during this period. Although subsequent public health communication campaigns over the past year changed their focus towards improving hand hygiene behaviour by recommending that hands be washed using water and soap for twenty seconds, there is still little research available on the current level of hand hygiene adequacy amongst the general public. Future research should focus on investigating the reasons for the failure to take heed of the public health ‘take-home’ messaging during a global pandemic, and it should also investigate effective interventions that can facilitate sustained, long-term behavioural change regarding adequate hand hygiene behaviour and compliance amongst the general public.

## Figures and Tables

**Figure 1 ijerph-18-06385-f001:**
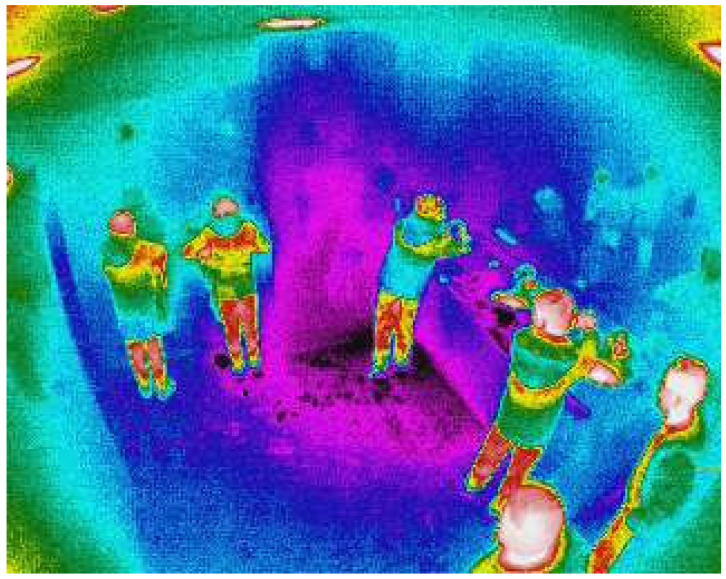
Infrared image of the male public restroom (wash-hand basin and hand dryer area only).

**Figure 2 ijerph-18-06385-f002:**
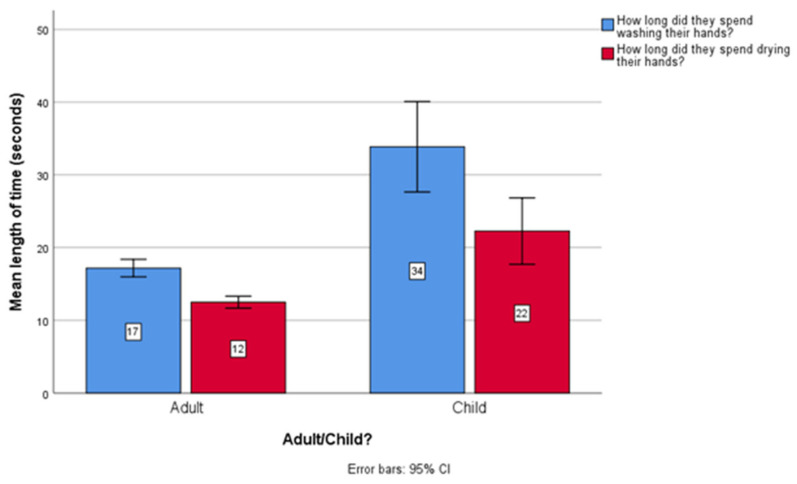
Average duration of time spent washing and drying hands for adults and children.

**Table 1 ijerph-18-06385-t001:** Hand hygiene compliance rate of the general public when using public restrooms during the COVID-19 pandemic.

Hand Hygiene (HH) Compliance Category	Total Hand Hygiene Compliance Rate during Observation Period % (*n)*				
	Total	Male	Female	Adult	Child
Adequate HH	17.07 (85)	19.69 (50)	14.34 (35)	14.35 (65)	44.44 (20)
Basic HH	62.65 (312)	57.48 (146)	68.03 (166)	64.02 (290)	48.89 (22)
Poor HH	13.05 (65)	12.99 (33)	13.11 (32)	13.69 (62)	6.67 (3)
Non-HH	7.23 (36)	9.84 (25)	4.51 (21)	7.95 (36)	0.00 (0)
Total	100.00 (498)	100.00 (254)	100.00 (244)	100.00 (453)	100.00 (45)

## Data Availability

The data presented in this study are available on request from the corresponding author. The data are not publicly available due to restrictions from the Funder.

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
