# Peer review of "An Evaluation of the Hand Hygiene Behaviour and Compliance of the General Public When Using Public Restrooms in Northern Ireland (NI) during the Initial Weeks of the Novel Coronavirus (COVID-19) Pandemic"

_ijerph, 2021, doi:10.3390/ijerph18126385_

Round 1
Reviewer 1 Report
The authors have addressed to all comments.
Author Response
The authors have addressed to all comments.
We have also repsonded to each of the second reviewer's comments.
Reviewer 2 Report
This is a very well written and easy to follow paper, with a thorough discussion of the findings. It makes an interesting contribution to the research given that most behavioural studies for COVID-19 have been self-report. There are obvious limitations to generalisability given the setting (public bathrooms during a period of high COVID-19 transmission) but I feel this is still an important contribution to the literature. I would recommend this paper for the IJERPH special issue on behavioural science during pandemics (https://www.mdpi.com/journal/ijerph/special_issues/Behavioural_Disease)
A couple of small comments below:
1. Introduction – so much has already been reported on hand hygiene (albeit self-reported) during COVID-19 and the introduction should reflect this. Currently it focuses too heavily on other circumstances outside of the pandemic. You could even argue that COVID research has mostly looked at self-reported data, as you do in the discussion. This will strengthen the rationale for the study.
2. Results: put number in brackets after %
I could see some revisions had been made in tracked changes - these revisions resolved any of the other comments I had.
Other notes:
1. It is interesting that children washed their hands for longer than adults - is this perhaps that parents/schools teach good hand hygiene to children but this wanes over time as we age and have less direct input from others?
2. People who were using public restrooms in March 2020 might have been less concerned about hygiene than the general population, as you mention at the very end of the discussion. I would potentially consider this point earlier in the discussion.
Author Response
This is a very well written and easy to follow paper, with a thorough discussion of the findings. It makes an interesting contribution to the research given that most behavioural studies for COVID-19 have been self-report. There are obvious limitations to generalisability given the setting (public bathrooms during a period of high COVID-19 transmission) but I feel this is still an important contribution to the literature. I would recommend this paper for the IJERPH special issue on behavioural science during pandemics
We will resubmit to this special edition to journal on the reviewer's recommendation.
A couple of small comments below:
- Introduction – so much has already been reported on hand hygiene (albeit self-reported) during COVID-19 and the introduction should reflect this. Currently it focuses too heavily on other circumstances outside of the pandemic. You could even argue that COVID research has mostly looked at self-reported data, as you do in the discussion. This will strengthen the rationale for the study.
- We added the following paragraph to the Introduction section:
"The common theme with many of these previous studies however is that they rely on using self-reporting methods when determining hand hygiene behaviour and compliance. The major issue with self-reporting methods is that they are often not a reliable indicator of true behaviour compared to other methods such as observation (17-18). Most of the research conducted during the course of the COVID-19 pandemic has relied on using self-reporting methods [31-33], many of which have focused solely on the availability of handwashing facilities, rather than the efficacy of hand hygiene behaviour within the general population."
- We added the following paragraph to the Introduction section:
2. Results: put number in brackets after %
The results in Table 1 have been changed to reflect this recommendation (see tracked changes).
I could see some revisions had been made in tracked changes - these revisions resolved any of the other comments I had.
Other notes:
- It is interesting that children washed their hands for longer than adults - is this perhaps that parents/schools teach good hand hygiene to children but this wanes over time as we age and have less direct input from others?
- This is something that we have considered in previous research looking specifically at handwashing knowledge, attitudes and practices in childcare settings (early years). Certainly there is evidence to suggest that early intervention and strong role models (such as parents, guardians and teachers) can heavily influence inherent behaviours like HH at this age. However, as children develop into teenagers, young adults and so on, there is a clear reduction in the level of adequate HH behaviour (as well as others that were reported to us around general hygiene e.g. brushing teeth, frequency of baths/showers) which is likely due to less input from key role models as you suggest. As such, there are little to no studies exploring HH behaviour in post-primary education and this is something we intend to investigate in the future to understand what differences in behaviour are by gender and age group in an attempt to understand why adequate HH behaviour reduces drastically by adulthood.
2. People who were using public restrooms in March 2020 might have been less concerned about hygiene than the general population, as you mention at the very end of the discussion. I would potentially consider this point earlier in the discussion.
We have moved this discussion point from the end to the start of the discussion (see tracked changes).
This manuscript is a resubmission of an earlier submission. The following is a list of the peer review reports and author responses from that submission.
Round 1
Reviewer 1 Report
The authors aimed to establish the level of hand hygiene behaviour and compliance of the general public during the initial weeks of the novel coronavirus (COVID-19) pandemic in a public restroom setting in Northern Ireland (NI).
The study is easy to follow and covers an interesting topic, but some issues should be improved before publication. Please check typos thorough the text.
Discussion section: Will be useful to the reader to add some interesting recent literature about the updates against COVID-19 outbreak and related tools to counteract the same (please see and briefly discuss: doi.org/10.3390/microorganisms9030525 ; doi: 10.12998/wjcc.v8.i18.3920;).
Conclusion Section: This paragraph required a general revision to eliminate redundant sentences and to add some "take-home message".
Author Response
The author's take onboard the reviewers comments and made the recommended changes.
Reviewer 2 Report
General Comments.
- This is an interesting paper and does give some valuable information on the prevalence of adequate hand washing during a pandemic. It also breaks the components down with regards to the time spent washing and time spent drying the hands.
- However, the paper itself should be reduced dramatically to the key elements. Every section could benefit from being made more concise and brief. In essence, this paper has only a few key findings:
- overall hand hygiene was not meeting the standard that is expected to prevent the spread of COVID-19,
- males are slightly worse than females but, while statistically significant, both need to improve,
- children, who do hand-hygiene, are more likely to do it properly.
- Table 1 has an error.
- I think the authors are pushing the interpretation of their study and have not given the reader sufficient evidence to support their statements. If they are citing studies (e.g. learned hand washing behaviour) then these need to be referenced. I think the statements on the communication strategy are pushing the limits of the data, particularly when no information on the public health communication for NI was provided.
Specific Comments.
- The paper could be made significantly shorter. The Introduction is far too long. For example, after the first sentence of the Introduction (lines 30 - 33), the authors could delete the text between lines 34 and 57. They could resume with the importance of hand hygiene since that is the focus of the paper.
- The number of significant figures needs to be reduced. For example, the authors write "Anderson and colleagues [20] established a hand hygiene compliance rate of 58.00%". They do not need the ".00". 58% is fine. This applies to all the other cited numbers as well as those in the tables.
- I think one photo image to show the infrared cameras is sufficient.
- Table 1 has an error in it. The numbers in the fifth column (under Total) do not add up to 498 and do not match with any of the other columns. These numbers need to be checked. In fact, I do not believe this column is even necessary since it should be a repeat of the second column and is thus unnecessary.
- The authors describe the results in a paragraph (lines 210 -228) and then repeat the same information in the table. This is unnecessary. My recommendation is to make one or two key points and refer to the table. They could indicate the p-values in the text but it might be better to indicate the confidence limits on some of the key estimates.
- The graphs on the time of hand washing are not necessary. I think the average time and percentage who met the recommended criteria for each (washing and drying) is sufficient.
- For the comparative T-tests, one sentence to say that there were no differences between males and females is sufficient. Readers do not need to know the details. The pattern of better washing among children is to be expected but still valuable to show.
- In the discussion, there needs to be a section on limitations of the study. The biggest one, from my perspective, is that the authors may have had a large selection bias. That is, those people who are more cautious and follow hygiene practices may be the ones who did not use public washrooms for whatever reason. The assumption is that the sample represents the general population and I think this needs to be acknowledged.
- In the discussion (line 288), the authors state that "The findings from this study also imply that self-reporting studies may not be a reliable indication of actual hand hygiene behaviour and compliance." I do not see the evidence to support this. I would like to see more to support this statement or have it removed.
- In the discussion (line 319), the authors state that "Gender differences in this study revealed that females had significantly better intention to practice hand hygiene compared to males". I think the term "significantly" refers to statistical significance. I do not believe the results were clinically significant. I think this section needs to be revised since the gender differences are minimal to say the least.
- The authors state that "The implications of this finding are profound in terms of evaluating the public response and public health communication strategy" (line 264). However, they have not presented any information on what the communication strategy was or contained. I feel that they have not given the readers sufficient evidence to support this statement. I would agree that the results are concerning in terms of general infection and control. However, I think their current statements about communication strategy, are pushing their data too far.
Author Response

(The authors gave the same response as above.)

Reviewer 3 Report
Dear authors, your manuscript has an important theme and worthy of study specially in the case of the COVID-19 outbreak. The study is well conduct.
The introduction is clear but too long. The methods part is well discribed and clear; the results part is enlightens us on the main results
Specific point:
Discussion : - line 323 please indicate reference "women generally more compliant" or change the sentence. I am drawing your attention to the healthcare workers are mostly women and the hand hygiene compliance are not good and variable.
Author Response

(The authors gave the same response as above.)
